# Ultra-high-throughput mapping of the chemical space of asymmetric catalysis enables accelerated reaction discovery

Wenjing Nie[1], Qiongqiong Wan[1], Jian Sun[1], Moran Chen[1], Ming Gao[1] & Suming Chen[1] ✉

The discovery of highly enantioselective catalysts and elucidating their generality face great challenges due to the complex multidimensional chemical space of asymmetric catalysis and inefficient screening methods. Here, we develop a general strategy for ultra-high-throughput mapping of the chemical space of asymmetric catalysis by escaping the time-consuming chiral chromatography separation. The ultrafast (~1000 reactions/day) and accurate (median error < ±1%) analysis of enantiomeric excess are achieved through the ion mobility-mass spectrometry combines with the diastereoisomerization strategy. A workflow for accelerated asymmetric reaction screening is established and verified by mapping the large-scale chemical space of more than 1600 reactions of α-asymmetric alkylation of aldehyde with organocatalysis and photocatalysis. Importantly, a class of high-enantioselectivity primary amine organocatalysts of 1,2-diphenylethane-1,2-diamine-based sulfonamides is discovered by the accelerated screening, and the mechanism for high-selectivity is demonstrated by computational chemistry. This study provides a practical and robust solution for large-scale screening and discovery of asymmetric reactions.

Asymmetric catalytic reactions have been the mainstays for the synthesis of chiral compounds[1–3], but the discovery of an efficient asymmetric reaction often requires extensive and laborious experimental efforts due to the complexity of structure-reactivity relationships of chiral catalytic systems[4,5]. The large number of possible combinations between catalysts, substrates, additives and reaction conditions constitutes a vast chemical space for asymmetric catalysis (Fig. 1a), and even modest structural changes to any or a few of these variables can have a profound effect on the experimental outcome[4–6]. The complexity of the structure-reactivity relationship makes it exceptionally difficult to screen catalysts and their generality, as well as to determine the optimal synthesis conditions for a specific product[7]. High-throughput screening (HTS) has become the critical approach for revealing the dark space of the reaction reactivities[7–12], however, the throughput of asymmetric reaction screening is bottlenecked by the determination of the enantiomeric excess (ee) values[13].

Although alternative approaches were developed for rapid assay of ee[9], such as the optical methods[14–17], [19]F-NMR[18], and mass-tagging with pseudo-enantiomers[8], the most commonly applied method for ee assay in asymmetric synthesis remains chiral chromatography due to its generality, direct ee determination, and reliability[7,19]. High-performance liquid chromatography (HPLC) with a chiral stationary phase, detected by ultraviolet (UV)-visible absorption or mass spectrometry (MS), allows accurate and direct readout of ee values[13,20]. Nevertheless, the long analysis time required for chromatographic separation greatly limits the efficiency of asymmetric reaction screening, even without considering the elusive stationary phase selection and pretreatment steps (Fig. 1b)[19]. This makes the current screening and optimization of asymmetric reactions hard to adequately consider as many combinations of conditions as other types of reactions, and makes it difficult to discover the best reaction system in a short period of time[21–26].

[1]The Institute for Advanced Studies, Wuhan University, Wuhan, Hubei 430072, China. ✉e-mail: sm.chen@whu.edu.cn

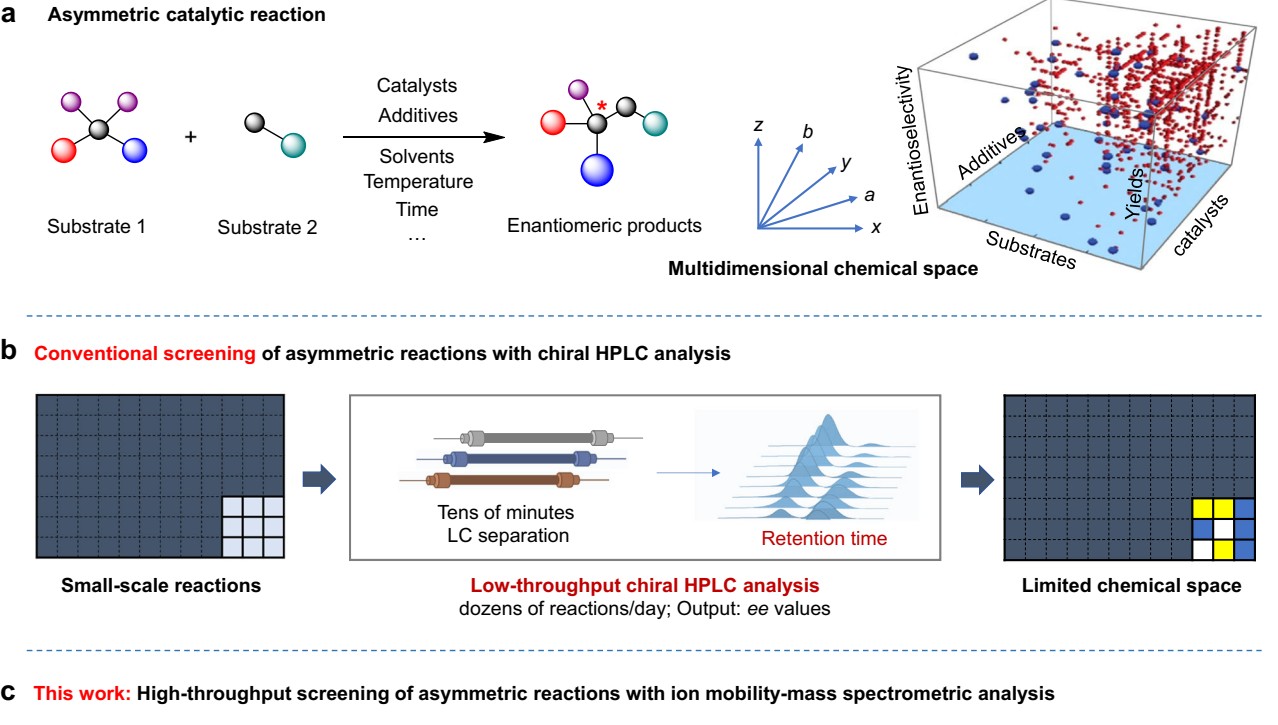

**a  Asymmetric catalytic reaction**

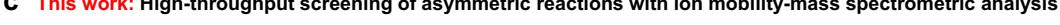

**b  Conventional screening** of asymmetric reactions with chiral HPLC analysis

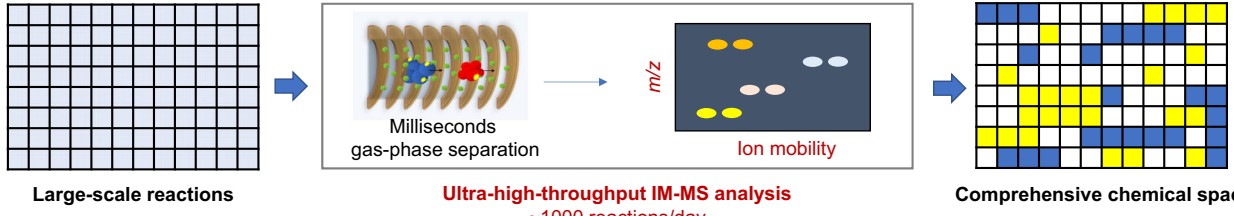

**c  This work:** High-throughput screening of asymmetric reactions with ion mobility-mass spectrometric analysis

**Fig. 1 | Schematic diagrams of asymmetric reaction screening. a** Asymmetric catalytic reaction and its multidimensional chemical space. **b** Procedures and limitation of conventional screening of asymmetric reactions by chiral HPLC analysis. **c** Procedures for proposed high-throughput asymmetric reaction screening with ultrafast ion mobility-mass spectrometry analysis, enabling elucidation of comprehensive chemical space.

In this study, we aim to develop a general ultra-HTS strategy for asymmetric reactions to expedite the mapping of the chemical space of asymmetric catalysis, as well as the discovery of enantioselective catalytic systems (Fig. 1c). The limitations of chiral chromatography such as time-consuming pre-purification and separation of enantiomeric products can be escaped by using ion mobility spectrometry (IMS) with MS detection. The millisecond separation of isomers in the gas phase without bias to the compound's functional groups allows for significant savings in analysis time per reaction, which affords an opportunity to rapidly survey the chemical reactivity landscape. We demonstrated the establishment of the IM-MS-based method for rapid assay of *ee* of asymmetric reactions at the speed of ~ 1000 reactions/day, and how the unique advances can be used to comprehensively reveal the masked chemical space of various catalysts and the generality to different substrates in asymmetric catalysis with ultra-high throughput. Especially, the accelerated discovery and optimization of primary amine-based organocatalysts were demonstrated.

## Results and discussion
### Development of the strategy for asymmetric reaction assay based on IM-MS
IM-MS is capable of isolating diastereomers on milliseconds scale and quantifying the diastereomeric ratio (*dr*) with high sensitivity (Supplementary Fig. 1)[27], but it hasn't been applied to the asymmetric reaction screening, probably due to its inability to directly analyze

enantiomers[28,29]. Given that the screening of catalytic systems usually proceeds with representative transformation using model substrates, we envision that if a derivatizable substrate is used in the reaction, the formed enantiomeric products could be transformed into the diastereomers through introducing a chiral resolving reagent by in situ derivatization (Fig. 2a)[29]. So, the enantioselectivity of the reaction could be rapidly evaluated by analyzing the ratio of the derivatized diastereomer products with IM-MS. This could become a general strategy for various asymmetric reactions that enables the highly efficient screening of the vast chemical space.

To this end, we first designed the chiral resolving reagents to achieve a good separation of the enantiomeric products, and chose the copper (I)-catalyzed azide-alkyne cycloaddition (CuAAC) for fast derivatization because of its high selectivity and equivalent conversion[30]. Due to the unique reactivity and high chemoselectivity between alkyl group and azide group, the effect of possible kinetic resolution between enantiomers could be greatly reduced. To further reduce the influence of kinetic resolution and ensure the accuracy of *ee* determination, we have specially designed the chiral resolving reagents by improving the distance between the azide group and the chiral center. Five of chiral reagents containing azide group were designed and synthesized, and the discrimination efficiency was evaluated for the standard of glycine (Gly) derivative Fmoc-propargyl-Gly-OH (**2a**) using trapped ion mobility spectrometry (TIMS)-MS (Supplementary Fig. 2)[27]. The racemic **2a** was successfully transformed

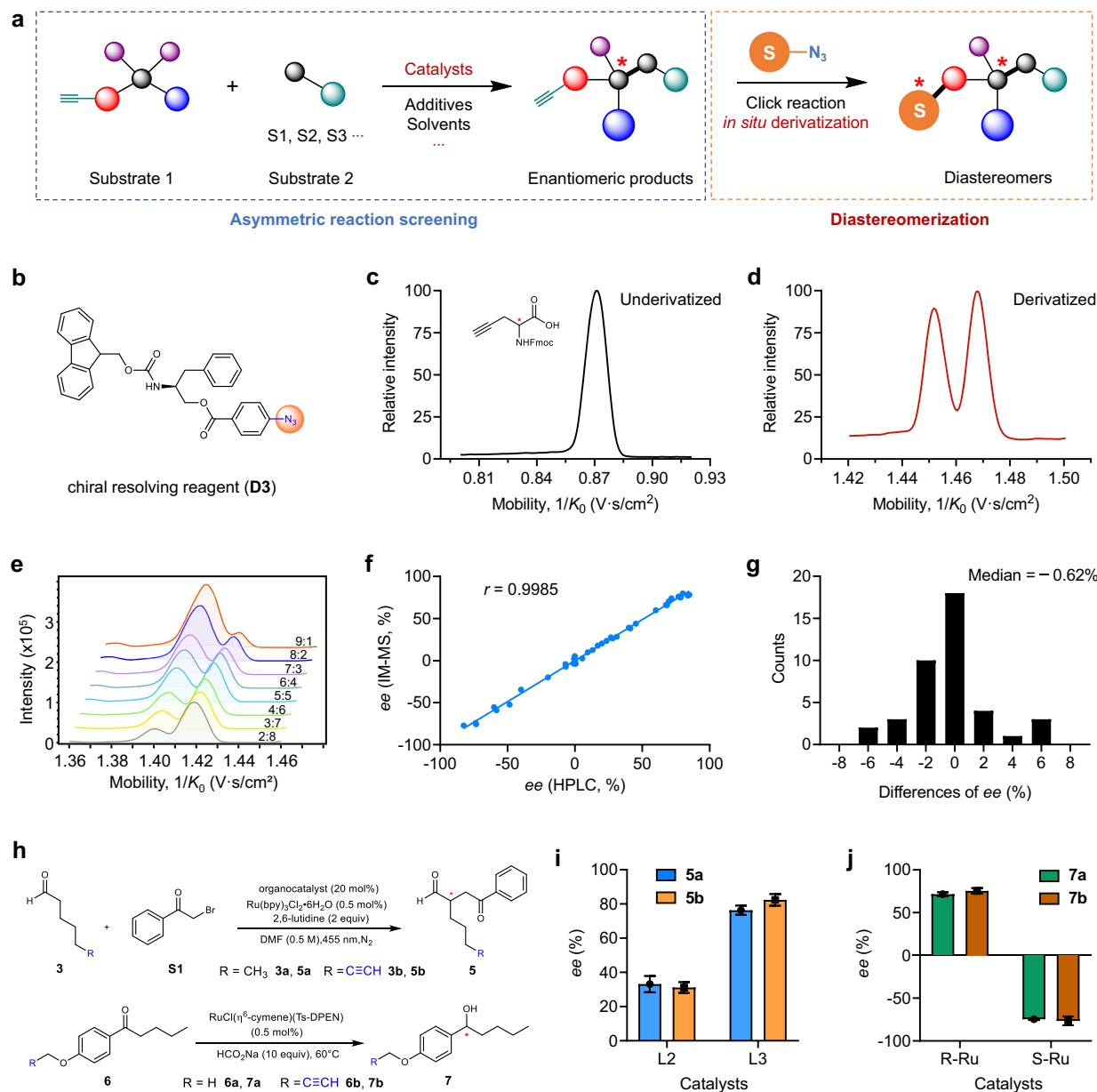

**Fig. 2 | Validation of the IM-MS-based strategy for the assay of asymmetric reactions. a** General strategy for the asymmetric reaction screening enabled by IM-MS analysis. **b** Structure of the chiral resolving reagent **D3**. **c** Extracted ion mobilogram (EIM) of the enantiomers of **2a**. **d** EIM of the derived diastereoisomers **D3**–**2a** from **2a**. **e** Stack of the EIMs of derived diastereoisomers from Fmoc-propargyl-Gly-OH (**2a**) in different ratios of *R*- and *S*-configuration using **D3** as chiral derivatization reagent. **f** Correlation of *ee* values determined by IM-MS with post-derivatization strategy versus those determined by chiral HPLC (*n* = 41). The Pearson correlation coefficient *r* = 0.9985. **g** Distribution of the relative errors of *ee* values determined by IM-MS (*vs.* HPLC). The median difference of *ee* is −0.62%. **h** Reactions of the direct asymmetric α-alkylation of aldehydes and Noyori asymmetric transfer hydrogenation. **i**, **j** Influence of alkynyl tag on the reaction enantioselectivity of (**i**) asymmetric alkylation of aldehydes and (**j**) Noyori asymmetric transfer hydrogenation. The *ee* values were determined by chiral HPLC, and the *ee* differences were labeled. Data are presented as mean values ± SD, *n* = 3 independent replicates.

into diastereomers by CuAAC reaction with chiral reagents, and the extracted ion mobilograms (EIMs) of different adduct ions of formed diastereomers revealed that the chiral reagents inspired by amino acid structure (**D1**, **D2** and **D3**) show more promising resolving power (Supplementary Fig. 2). The reagent **D3** ((*S*)-2-((((9*H*-fluoren-9-yl) methoxy)carbonyl) amino)-3-phenylpropyl 4-azidobenzoate, Fig. 2b) was ultimately chosen for subsequent experiments because the sodium ion adducts of the derivatized diastereoisomers have the highest separation resolution ($R_s$ = 1.04) (Fig. 2c, d). In addition, the molar ratio of the **2a** enantiomers was found to be linearly correlated with the peak area ratio of the derived diastereoisomers

(Supplementary Fig. 2 and Fig. 2e). The successful resolving of chiral aldehyde and alcohols with **D3** (Supplementary Fig. 3) was also achieved, suggesting the broad applicability of this strategy to different reactions. Note that the baseline separation of enantiomeric pairs is not necessary, and the accurate quantification of the peak areas could be realized by using the curve-fitting software for processing the extracted ion mobilograms, which was confirmed by further quantitative experiments of the proposed strategy using chiral reagent derivatization and IM-MS. We used 41 pair of enantiomer mixtures with different *ee* values to compare the quantitative results between chiral HPLC and IM-MS (Fig. 2f, g, Supplementary Fig. 4, and

Supplementary Table 1). The results show that the *ee* values of the enantiomers could be directly determined based on the peak areas of the derived diastereoisomers in EIMs, and the values were highly consistent with those measured by chiral HPLC (Pearson correlation coefficient *r* = 0.9985, Fig. 2f). The median difference of *ee* was only −0.62% (Fig. 2g), indicating that the method can be used to determine the *ee* values of the enantiomeric products accurately without using standards and calibration curves. In addition, the influence of alkynyl tag on the enantioselectivity was also investigated by using two asymmetric reactions: the direct asymmetric α-alkylation of aldehydes[31] and Noyori asymmetric transfer hydrogenation reaction[32] (Fig. 2h). The results show that the enantioselectivities of these reactions are much similar when the alkynyl-containing substrates or those without alkynyl were used (*ee* differences 2%-6%, Fig. 2i, j). Overall, the above results demonstrate the feasibility of using this post-derivatization strategy and IM-MS to rapidly and reliably determine the *ee* of asymmetric reactions.

## High-throughput mapping the chemical space of asymmetric catalytic reactions

Having developed the strategy and method for rapid analyzing the enantiomeric products, we dedicate to establish a compete workflow for the high-throughput screening and mapping the chemical space of asymmetric catalytic reactions. The direct asymmetric α-alkylation of aldehydes merged with photoredox catalysis and organocatalysis (Fig. 3a) was chosen as a benchmark study[31] due to its broad value in asymmetric synthesis. The multiple variables and dimensions of this reaction also makes it a good candidate for demonstrating the performance of the screening platform.

To initiate the reaction, a home-made photochemical reaction chamber compatible with 96-well plate microreactor was designed (Supplementary Fig. 6). To survey the reactivities and enantioselectivities, hept-6-ynal (**3b**) was used as the derivatizable substrate, and 10 bromoacetophenone substrates (**S1-S10**), 11 secondary amine organocatalysts (**L1-L11**), and 13 photocatalysts (**P1-P13**), including 6 transition metal catalysts and 7 dye catalysts, Supplementary Fig. 7) were selected with orthogonal combination for totally 1430 reactions (Fig. 3a). We focused on the investigation of organocatalysts and the synergic catalytic effect with photocatalysts. The involvement of different bromoacetophenone substrates could examine the generality of the catalysts. *N*,*N*-dimethylformamide (DMF) and 2,6-lutidine were chosen as the general solvent and the base additive, respectively, for the whole reactions[31]. These combinations would encompass regions of chemical space that are difficult to be explored by conventional methodologies.

The assessment of the enantioselectivity is critical to the screening of asymmetric reactions. We have demonstrated the high-efficiency of the diastereoisomerization and the subsequent IM-MS analysis for the rapid *ee* assay of enantiomeric products. In this case, the derivatization using highly-polarity chiral reagents can not only resolve the enantiomeric isomers, but also solve the inherent challenge for the analysis of organic reactions by electrospray ionization MS, i.e., the poor ionization efficiency of many organic compounds with low-polarity. After the short period (10 min) of derivatization with chiral resolving reagent **D3** via CuAAC reaction, the solutions in 96-well plate were automatically subjected to IM-MS analysis with a well-plate autosampler. The average time cost of derivatization is only a few seconds for one reaction and has little impact on throughput when screening a large number of reactions. The total analytical time for one reaction is only about 1.5 min including sample injection, IM-MS analysis, and loop cleaning. So, the analyses of the whole 1430 reactions could be done within 40 h. The representative EIMs (Fig. 3c and Supplementary Fig. 8) of the screened reactions show the *ee* of each reaction could be easily obtained in a straightforward way. These results validated the practical applicability of this highly efficient screening platform for asymmetric reactions.

## Multidimensional chemical space of the photocatalytic asymmetric alkylation reaction

The analytical results of these reactions present the visualization of the large chemical space of the photocatalytic asymmetric alkylation reaction of aldehydes (Fig. 3b and Supplementary Fig. 9), which provides a visually straightforward manner for identifying correlations in performance between different variables. Notably, this large-scale enantioselectivity data composes 1430 *ee* values from −78% to 90%, and the statistical analyses of these data could provide a deeper insight into the generality of the catalysts and substrates (Fig. 4). Given that the median is the most informative measure of central tendency, which reflects the distribution of the data set and is not affected by extreme values in the distribution, we then calculated the median *ee* values for different reactions using specific catalysts or substrates to assess the level of generality for each catalyst. For the organocatalysts, **L2** (0.46), **L3** (0.55), **L4** (0.56) and **L10** (0.49) were observed to have high median *ee* values among the chiral amine catalysts (Fig. 4a), which indicate general high enantioselectivity for most of the bromoacetophenone substrates and photocatalysts. Among these catalysts, the chiral products obtained with **L2** and **L3** are mainly *R*-configuration, while those obtained with **L4** and **L10** are mainly *S*-configuration. Unlike organocatalysts, the photocatalysts did not exhibit clustered enantioselectivities (Fig. 4b), and the *ee* values are divergent for each catalyst. Similarly, the substituents of different substrates didn't show preferred enantioselectivities (Fig. 4c). These results reveal that the enantioselectivity of the direct asymmetric alkylation of aldehydes mainly depends on the structure of organocatalysts.

Besides the enantioselectivities, the IM-MS-based strategy also allows the approximately assessment of the relative yields of the reactions. By adding another alkyne-containing compound 1-phenylprop-2-yn-1-ol (**4**) as an internal standard, the relative MS yields of different reaction systems could be compared by calculating the relative peak areas of ions ($A_R = A_{product}/A_{internal\ standard}$)[33] between the derivatives from the reaction products and internal standard while determining the *ee* values. In this case, we used one internal standard to correct for products with different substituents, and the ionization efficiencies may differ slightly between products. To minimize the variation, the protonated MS peaks was used for quantification. This is because both the product and the internal standard derived from the click reaction contain a triazole structure, which has a high and similar protonation efficiency in different compounds. Therefore, the ionization difference can be minimized to increase the accuracy of quantification. As shown in Fig. 4d, e, the organo- and photocatalysts were observed to have great influence on the formation of the product in this reaction. Especially, the transition metal complex photocatalysts can obtain generally higher yields than dye photocatalysts. In addition, the substrates with electron-withdrawing group on the aromatic ring (**S6-S10**) can obtain relatively higher yields than others (Fig. 4f). The dimension of relative yield makes the screened results more informative. The deep insights obtained from the large-scale chemical space mapping demonstrated the great advantages of the developed ultra-HTS method.

## High-efficiency screening of primary amine-based enantioselective organocatalysts

Although great progress has been made in asymmetric organocatalysis, catalysts capable of catalyzing the enantioselective conversion of aldehydes to yield highly selective chiral products are still mainly limited to secondary amines, such as **L2**, **L3** and **L4** mentioned before. Whether other types of highly enantioselective catalysts (e.g., primary amines) exist is still unknown. Primary amine-based catalysts were used for enantioselective synthesis in some studies[34], but they were rarely effective for asymmetric α-alkylation reactions of simple aldehydes[35]. Having established and verified the high-throughput screening strategy and platform, we next aim to discover a type of efficient primary amine-based organocatalysts for asymmetric α-alkylation of aldehydes.

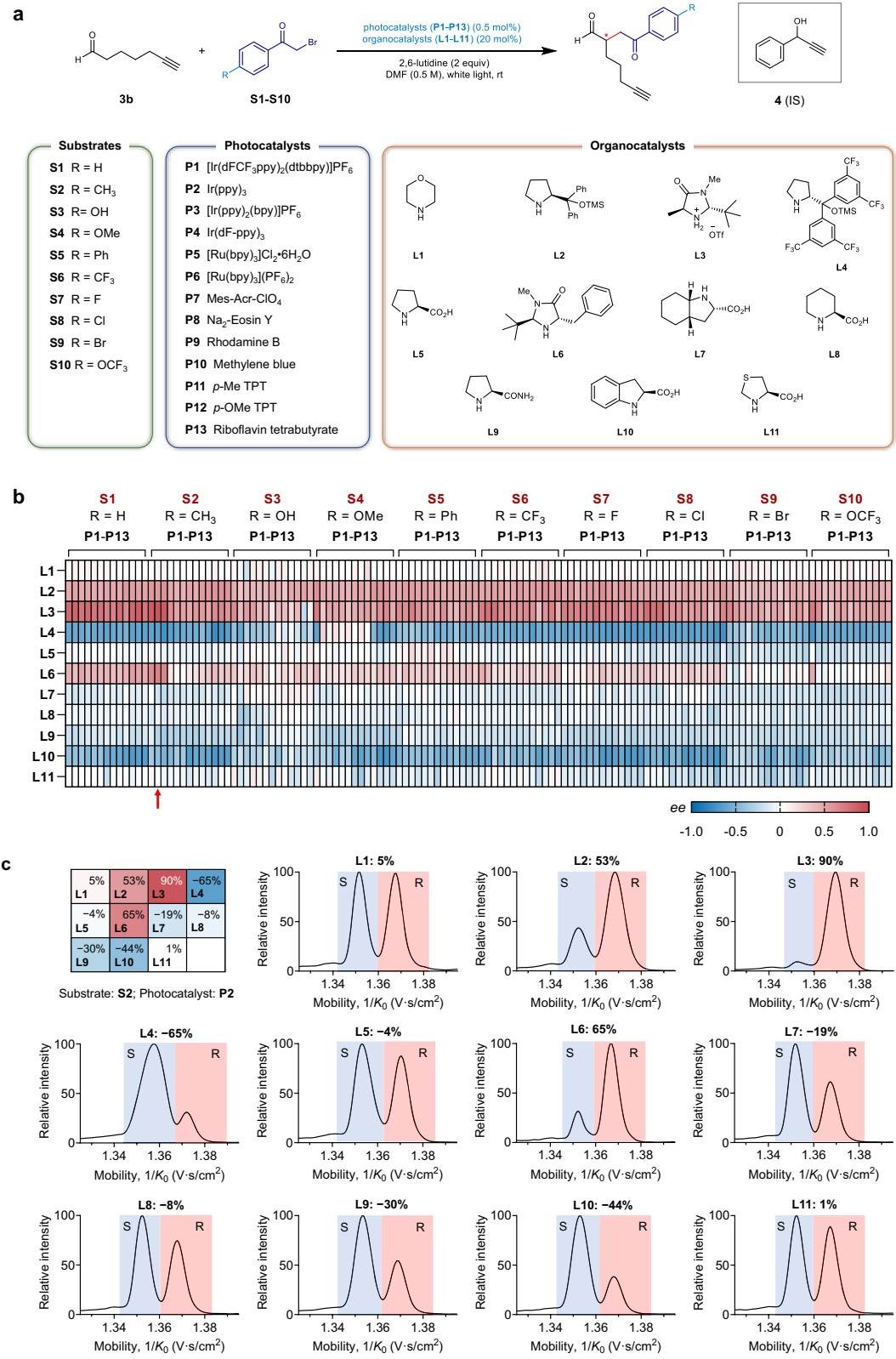

**Fig. 3 | Combinatorial screening of the direct α-asymmetric alkylation of aldehydes merged with photoredox catalysis and organocatalysis. a** Chemical reaction formula and the structures/names of the 10 bromoacetophenone substrates, 11 organocatalysts, and 13 photocatalysts. **b** Heatmap visualization of the *ee* values of the 1430 reactions determined by the IM-MS. **c** Eleven of selected extracted ion mobilograms (EIMs) for the screened reactions with substrate **S2**, photocatalyst **P2** and different organocatalysts.

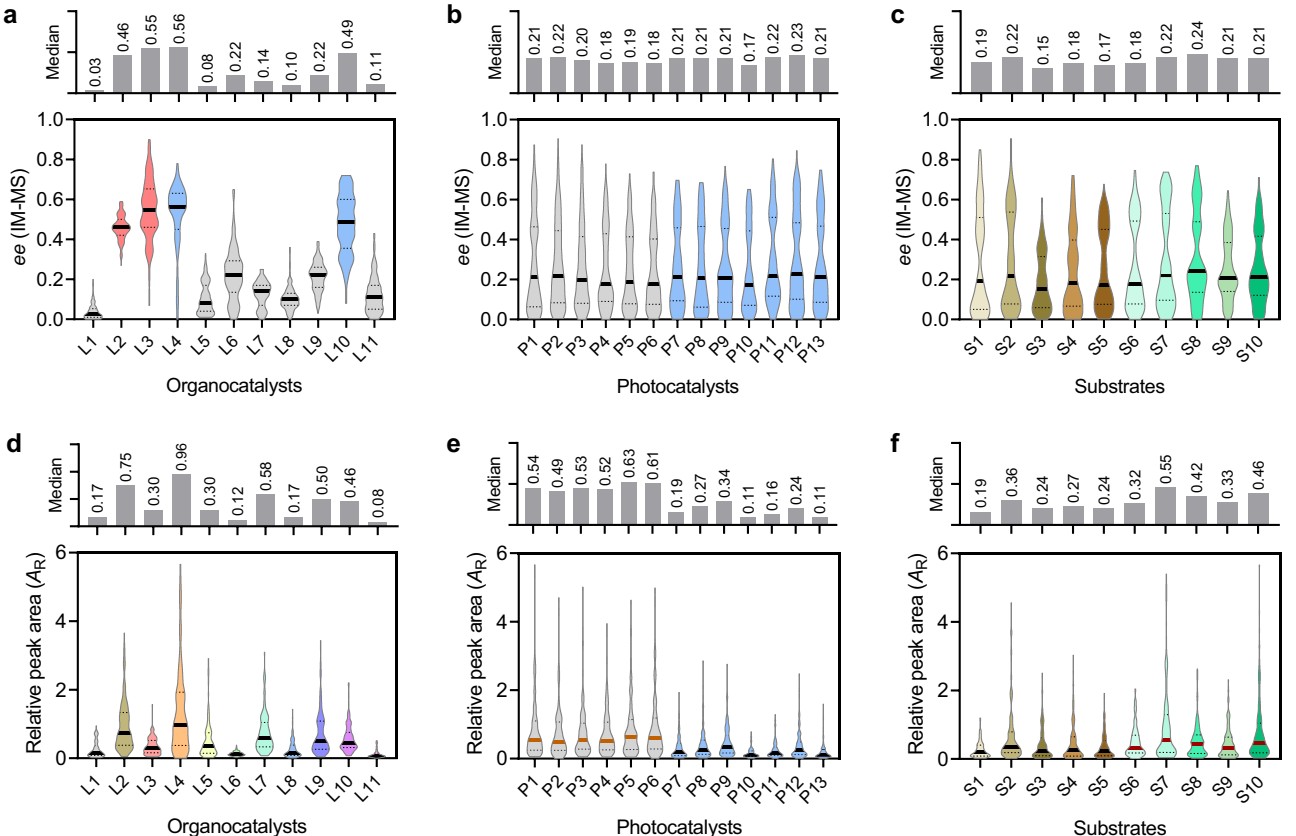

**Fig. 4 | Statistical analysis of the 1430 screened asymmetric reactions from the perspective of organocatalysts, photocatalysts and substrates. a–c** Violin plots of *ee* values with different (**a**) organocatalysts, (**b**) photocatalysts and (**c**) substrates. **d–f** Violin plots of relative MS yields with different (**d**) organocatalysts, (**e**) photocatalysts and (**f**) substrates. The median is represented with the middle solid line, and the interquartile ranges are represented by the outer dashed lines. The bar chart at the top of each panel shows the comparison of medians.

Based on the previous proline-based secondary amine catalytic system, we conducted preliminary screening of primary amine-based organocatalysts with common chiral L-amino acids, their analogues and some building blocks (Fig. 5). The representative reaction system (Fig. 2h) with aldehyde **3b** and α-bromocarbonyl **S1** and photocatalyst Ru(bpy)₃Cl₂·6H₂O (**P5**) were chosen to test the performances of the chiral organocatalysts. The *ee* values of the enantiomeric products of these 32 reactions could be rapidly determined by IM-MS-based method within 1 hour. As shown in Fig. 5, most of the unmodified amino acids show low enantioselectivity, while L-threonine (**A13**, −29% *ee*) and L-serine (**A16**, −27% *ee*) provided enantioselective transformation of aldehydes, the modified structures of threonine (**A15**, −5% *ee*) and serine (**A18**, −6% *ee*) with large steric hindrance groups instead yielded much lower enantioselectivity. Interestingly, the reaction shows an improved enantioselectivity with ethyl L-serine (**A17**, −48% *ee*), and the alkylamine compound (*S*)-2-amino-3-phenylpropan-1-ol (**A22**, 40% *ee*), (1*R*,2*S*)-2-amino-1,2-diphenylethan-1-ol (**A23**, −62% *ee*) and (1*S*,2*S*)-1,2-diphenylethane-1,2-diamine (**A24**, 41% *ee*) also exhibited moderate enantioselectivity. To our delight, we found the structural analogs of 1,2-diphenylethane-1,2-diamine (DPEN), **A28** (75% *ee*) and **A29** (64% *ee*), exhibit high enantioselectivity, although other compounds with the structure 1,2-diamine have very low enantioselectivity (**A25-A27** and **A36-A38**).

**Verification of highly enantioselective catalysts containing DPEN-based sulfonamides**

The discovery of the high-enantioselectivity of DPEN-based sulfonamides (**A28** and **A29**) inspires us to further verify this type of primary amines in catalyzing the asymmetric α-alkylation of aldehydes. A total

of 8 analogues (**A28-A35**) were selected and synthesized for the following studies (Fig. 6a). Prior to the screening, the reaction conditions were optimized in a high throughput mode by using substrates **3b** and **S1**, photocatalyst **P5** and organocatalyst **A30** (Supplementary Fig. 10). We firstly evaluated five solvents, four additives and five forms of **A30** and found that the reaction exhibited higher enantioselectivity in amide solvents like DMF and DMA, while the target product was trace in DMSO. The additives also had a significant effect on the enantioselectivity of the product and the combination of DMF and 2,6-lutidine was found to be optimal for the catalysis of **A30** (83% *ee*). Notably, we found that the equivalence ratio of the organocatalyst to the substrate had a large effect on the enantioselectivity of the reaction (Supplementary Fig. 11). The 30 mol% organocatalyst was ultimately used for subsequent catalyst screening.

To investigate the synergic catalysis of organocatalysts with photocatalysts, we evaluated the catalytic effect of the combination of 8 primary amines and 13 photocatalysts on the asymmetric α-alkylation reaction of **S1** (Fig. 6a). Screening this set of variables would provide data for 104 reactions. As shown in Fig. 6b, the DPEN-based sulfonamides (**A28-A35**) show overall good enantioselectivity in combination with different photocatalysts. The combination of **A30** and **P5** can obtain 88% *ee* for the asymmetric α-alkylation of **3b** using **S1** as substrate. Compared with the other structures, **A29** and **A33** exhibit relatively low enantioselectivity, suggesting that the attachment of electron-rich aromatic ring to the sulfonamide group, is more favorable to the enantioselectivity of the reaction.

The high-throughput screening platform also offers convenience to explore the generality of the catalyst, and **A30** was used as an example to explore its generality to different substrates and

**Fig. 5 | Screening of primary amine-based enantioselective organocatalysts.** Heatmap visualization of the *ee* values of the reactions catalyzed with different primary amine-based organocatalysts. The *ee* values were determined by the developed IM-MS-based method. Gray boxes for **A8, A10** and **A11** indicate that the product was not acquired.

photocatalysts. An orthogonal combination of 91 reactions consisting of 7 substrates and 13 photocatalysts was performed (Fig. 6c). The results showed that most of the reactions exhibit high enantioselectivity, with 4 substrate-photocatalyst combinations (**S6-P5, S6-P5, S7-P2** and **S8-P1**) obtaining products with *ee* values above 90%, indicating that the developed DPEN-based sulfonamides have great potential for asymmetric catalysis. This kind of primary amine-based organocatalysts would enrich the chiral amine toolbox in asymmetric organocatalysis.

The superior enantioselectivity of sulfonamide-containing organocatalysts encourages us to explore the underlying mechanism. We first identified the enamine intermediate **11** (*m/z* 445.1939, error −1.1 ppm) and **12** (*m/z* 383.1787, error −0.3 ppm) during the catalytic process by high-resolution MS through mixing the aldehyde **3b** with **A28** and **A29** in CH₃CN (Figs. 6d, e). Then, density functional theory (DFT) was applied to calculate the conformation of the enamine intermediate **11** and **12** (Fig. 6f). The result indicates the dual hydrogen bonds[36] (N−H···N and N−H···O) between sulfamide group and enamine group could be formed, which may assist the formation of *Si* face exposed for the enantioselective radical addition. Notably, the N−H···O hydrogen bonds belong to the strong interaction due to the relative short donor-acceptor distances of 2.22 and 2.25 Å in **11** and **12**, respectively. This interaction may explain the specific role of the sulfonamide group in the catalysts for the high enantioselectivity.

In summary, a general methodology for the ultra-high-throughput *ee* determination and asymmetric reaction screening was developed by combing the diastereoisomerization strategy and IM-MS analysis in this study. Compare with traditional methods, the workflow proposed in this study trades a convenient post-derivatization for the ability to analyze the reaction outcomes with ultra-high-throughput and multidimensional information. As an example, the comprehensive mapping of the chemical space of asymmetric α-alkylation reactions was achieved, which provides insights into the performance and generality of organocatalysts and photocatalysts and their synergistic effects. More importantly, a class of primary amine organocatalysts of DPEN-based sulfonamides was discovered, and their high generality for different transformations was validated. The developed method will be quite helpful for the ultra-fast characterization and discovery of the promising asymmetric catalytic systems.

## Methods

### IM-MS analyses of diastereomers and derivatized enantiomers

To explore the discrimination power of IM-MS for diastereomers, chiral phosphoramidite ligands **1a** (Supplementary Fig. 1) were selected and subjected to IM-MS analysis. Firstly, (*S,S,S*)-**1a** (5 mM) and (*S,R,R*)-**1a** (5 mM) in MeOH were mixed to form the solutions of diastereomers with different ratios. Then, KCl was added as the source of K⁺ and the finally concentration of KCl was 100 µM. The final solutions were subjected to IM-MS analysis injected by LC autosampler with 100% MeOH as mobile phase (flow rate: 0.3 mL/min). The analysis was operated in "ultra" mode at a range of 1.09-1.19 V·s/cm² with lock duty cycle to 100%, and the general settings were listed as following: capillary voltage 3.5 kV, end plate offset 500 V, nebulizer gas 2.0 bar, dry gas 10 L/min, dry temperature 200 °C.

The IM-MS analysis for alkynyl-contained enantiomers was operated in "custom" mode at a certain mobility range (depends on the

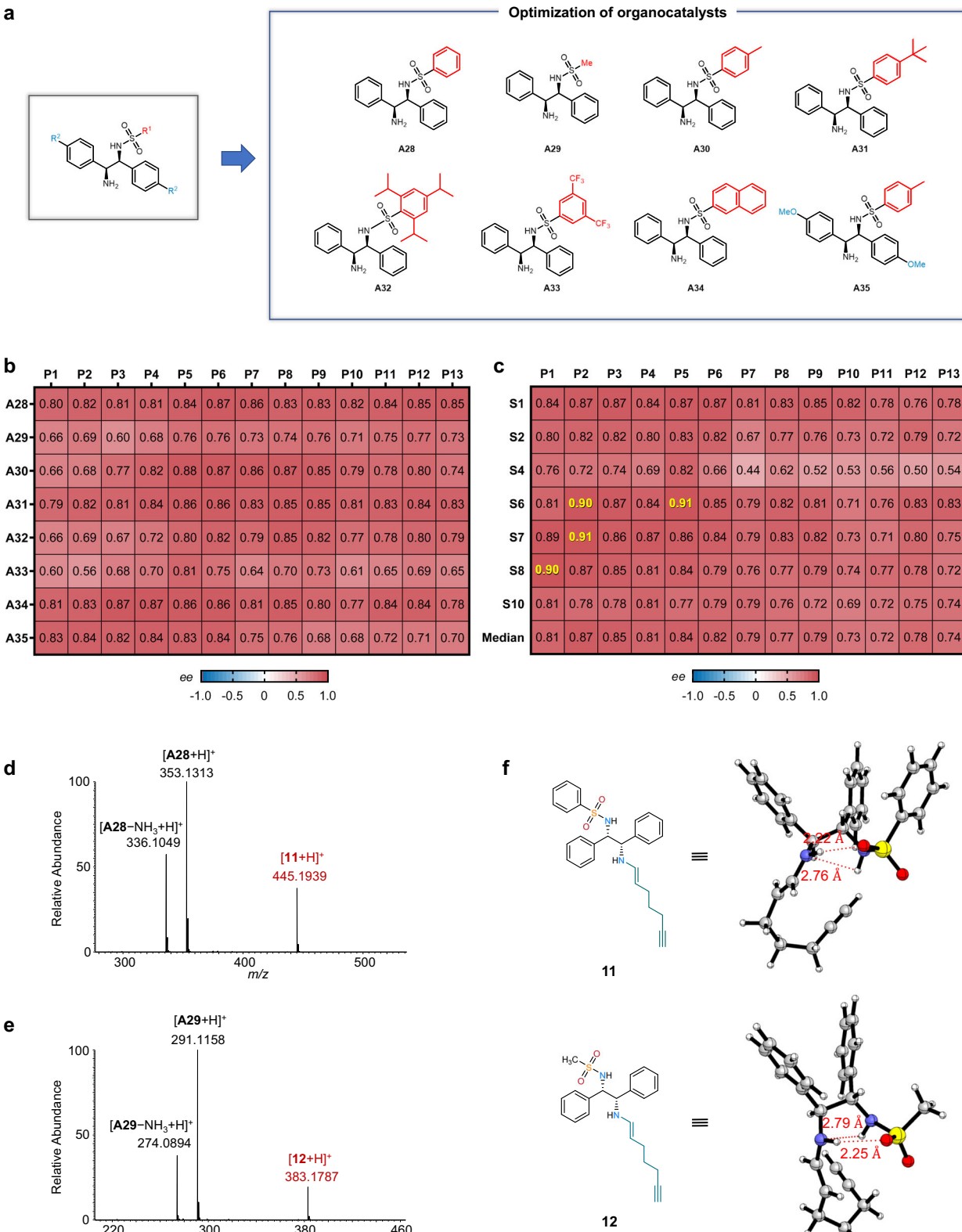

**Fig. 6 | Development of the DPEN-based sulfonamides organocatalysts for asymmetric α-alkylation reactions. a** Optimization of the structure of the DPEN-based sulfonamides organocatalysts. **b** Enantioselectivity of DPEN-based sulfonamide organocatalysts and photocatalysts for synergistic catalysis. Reactions were performed by using the substrates **3b** and **S1**. **c** Generality of the organocatalyst **A30** in the asymmetric α-alkylation reactions. Aldehyde **3b** was used as the substrate, and the photocatalyst and organocatalyst equivalents were 0.5 mol% and 30 mol%, respectively. **d**, **e** The MS detection of enamine intermediate (**d**) **11** and (**e**) **12** with [M + H]+ ions. **f** Conformations of enamine intermediates **11** and **12** obtained from DFT calculation. The dual hydrogen bonds interactions between sulfamide group and enamine group may account for the high enantioselectivity of the catalysts. N atoms were colored with blue; O atoms were colored with red, and S atoms were colored with yellow.

mobility of specific diastereomer adduct ions) with lock duty cycle to 100%, ramp time: 900 ms and the general settings were listed as following: capillary voltage 3.5 kV, end plate offset 500 V, nebulizer gas 2.0 bar and dry gas 10 L/min (nebulizer gas 0.3 bar and dry gas 3.5 L/min when the syringe pump was used for injection), dry temperature 200 °C, $m/z$ scan range: 150-1350 Da. The sample solutions were injected by LC autosampler with 90% MeOH and 10% $H_2O$ as mobile phase (flow rate: 0.3 mL/min).

For the analysis of the screened asymmetric reactions, 100 $\mu M$ NaCl was added to the mobile phase MeOH/$H_2O$ (9/1, v/v) to increase the abundance of formed sodium adducts ions. The analytical time for *ee* determination of each asymmetric reaction by timsTOF Pro mass spectrometer is about 1.5 min, including 50 s for looping cleaning, sample injection and 30 s for IM-MS analysis. The timsTOF Pro mass spectrometer was daily calibrated using the Agilent ESI low-concentration Tuning mix. The analysis was operated in "custom" mode with lock duty cycle to 100%, ramp time: 900 ms, set $1/K_0$ end for accumulation to 1.90 V·s/cm², and the general settings were listed as following: capillary voltage 3.5 kV, end plate offset 500 V, nebulizer gas 2.0 bar, dry gas 10 L/min, dry temperature 200 °C, $m/z$ scan range: 100-1000 Da, The RF funnel 1 and 2 were set at 250 Vpp, multipole RF at 200 Vpp and the deflection delta voltage was 80 V. The quadrupole ion energy was 5 eV, low mass was 150 Da, transfer time 65 $\mu s$ and pre-pulse storage time 5 $\mu s$. The mobility range was set as follows: 1.30-1.40 V·s/cm² for **S1**; 1.32-1.42 V·s/cm² for **S2, S4, S6** and **S8**; 1.31-1.37 V·s/cm² for **S3**; 1.36-1.46 V·s/cm² for **S5**; 1.28-1.38 V·s/cm² for **S7**; 1.33-1.43 V·s/cm² for **S9**; 1.35-1.45 V·s/cm² for **S10**. The EIM of each reaction product was analyzed by the corresponding formed diastereomers with [M + Na]⁺ adduct ion, such as the enantioselectivity analysis of reactions operated with **S1** was refer to the EIMs of 769.2977 Da.

### Derivatization procedure for alkynyl-contained enantiomers and reaction mixtures

The general procedure for the derivatization of alkynyl-contained enantiomers began with the solution preparation of enantiomers (10 mM in ACN), chiral resolving regent (10 mM in DCM), and CuI with *N,N*-diisopropylethylamine (DIPEA) as the ligand (10 mM in ACN). Then, the three components (100 $\mu$L/EA) were mixed in a 1.5 mL centrifuge tube to conduct the CuAAC derivatization reaction. The reaction was proceeded in an ultrasonic water bath for 10 min at room temperature, then the tube was centrifugated to make the copper catalyst deposited. Finally, the supernatant was collected and diluted to 100 $\mu$M with ACN before the IM-MS analysis.

For the derivatization of the enantiomeric products for the high-throughput screening of asymmetric reactions, 4 $\mu$L mixture of each reacted solution was collected and diluted with 36 $\mu$L ACN in another 96-well plate (450 $\mu$L Nunc U96 Microwell, Thermo Scientific). Then, 40 $\mu$L/EA of chiral resolving regent (150 mM in DCM), CuI with DIPEA (400 mM in ACN), and internal standard 1-phenylprop-2-yn-1-ol (**4**) (25 mM in ACN) were added to each well. Subsequently, the 96-well plates were sealed with plate covers to avoid the volatilization of ACN, and suspended in an ultrasonic water bath for 10 min at room temperature. After the reaction, 10 $\mu$L of supernatant for each reaction was transferred to another 96-well plate and diluted with 190 $\mu$L ACN before the IM-MS analysis. The concentration of the product after derivatization is 625 $\mu$M if the reaction yield is 100%. All liquid transfer processes were handled by an automated liquid handing system (VERSA 110, Aurora Biomed, Canada).

### Combinatorial screening of the direct asymmetric α-alkylation of aldehydes with photoredox catalysis and organocatalysis

The high-throughput experiments of asymmetric reactions were initiated with hept-6-ynal (**3a**), bromoacetophenone substrates **S1-S10**, organocatalysts **L1-L11**, photocatalysts **P1-P13** (see Fig. 3a & Supplementary Fig. 5). The selected model reaction was operated in

0.5 M, so we prepared the reaction components stock solutions as following: **S1-S10** (2 M in DMF, 5 mL), **L1-L11** (400 mM in DMF, 5 mL), **P1-P13** (10 mM in DMF, 5 mL), **3a** and 2,6-lutidine mixture (4 M in DMF, 40 mL). Then combinatorial mixed these components with equal 25 $\mu$L to 96-well plates by the automated liquid handing system. After placing the 96-well plates in ultrasonic water bath for 10 seconds to mix the reaction uniformly, each 96-well plate was covered with an optical glass to minimize the volatilization of solvent and components, and two 96-well plates were placed side by side into a home-made aluminum alloy container (see Supplementary Fig. 6). A transparent acrylic top layer was fixed to the container with 12 flange bolts, then the container was degassed with an oil pump for three times and refilled with nitrogen and the container was connected with a nitrogen balloon to ensure a nitrogen atmosphere for the reaction. To initiate the photochemical reactions, a 100 W LED compact fluorescent lamp (CFL) was placed approximately 8 cm to the top layer, and the container with ice-cooling was placed on a shaker to reduce the impact of mass transfer and heat transfer to reactions operated in 96-well plate without magnetic stirring. After irradiated for 8 h, the 96-well plates were transferred to the automated liquid handling system (Supplementary Fig. 6) for the post-derivatization and dilution for subsequent IM-MS analysis and the relative MS yield determination. The relative MS yield was determined by an Orbitrap Elite™ mass spectrometer with the following settings: FTMS positive mode, spray voltage: 3.8 kV, source heater temp: 350 °C, sheath gas flow rate: 40, aux gas flow rate: 10, $m/z$ scan ranges: 400–900, resolution: 120,000, acquire time: 0.8 min.

### The detection of enamine intermediates and the computational method

The enamine intermediate **11** was formed with 10 $\mu$L **3b** (10 mM stock solution in ACN) and 10 $\mu$L **A28** (10 mM stock solution in ACN), and the mixture was diluted with ACN and the final concentration was 50 $\mu$M for the MS analysis by Orbitrap Elite™ mass spectrometer. Accordingly, the enamine intermediate **12** was detected by mixing **3b** and **A29**. All density functional theory (DFT) calculations were carried out using the Gaussian 16 software Revision A.03. The geometries were optimized using the B3LYP-GD3BJ functional with a basis set of def-TZVP for all atoms. Vibrational frequency calculations were performed for all the stationary points to confirm if each optimized structure is a local minimum or a transition state structure.

### Data analysis

Thermo Xcalibur (version 4.1) software was used for the analysis of relative MS yields of asymmetric by extracting formed diastereomers target [M + H]⁺ and calculating their peak areas ratio from the mass spectrometric data generated by Orbitrap Elite mass spectrometer. Compass Data Analysis 6.2 (Bruker Daltonics, Germany) software was used for extracting the target ion mobilogram and integrating the target compounds manually. When the extracted ion mobilogram peaks of the formed diastereomers overlapped, the OriginPro 9 software was used for the peak fitting. The peak fitting procedure was used only for the reactions with **S3** as substrate, the others were processed directly using Compass Data Analysis 6.2. Here are the parameter settings and procedures for the peak fitting of the reactions with **S3** as substrate: 1) Export the raw data of the extracted ion mobilogram as an. xy file using the Bruker Compass Data Analysis 6.2 software; 2) Open the text file with OriginPro 9 software and plot the selected data as a line graph; 3) Select the line graph, click in sequence on Analysis, Peak and Baseline, Peak Analyzer, Open dialog theme. Peak fitting parameter settings: Recalculate: Manual, Goal: Fit peaks (Pro), Baseline Mode: Constant with Minimum, Peak type: Gaussian, find peaks by height and fit peaks with fitting the two peak area and full peak width at half-maximum refer to the racemic result, and fit converged with

Chi-Sqr tolerance value of 1E-6 was reached. The two fitted peak areas used for *ee* calculation were obtained in the results log.

## Data availability

The mass spectrometric data generated in this study have been deposited in ProteomeXchange Consortium (https://www.iprox.org/). Project ID: IPX0007235000. Source data are provided with this paper and also in Figshare (https://doi.org/10.6084/m9.figshare.22826312). All other data are available from the corresponding author upon request. Source data are provided with this paper.

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

## Acknowledgements

This work was financially supported by the National Natural Science Foundation of China [22074111 (S.M.C.), 22004093 (Q.Q.W.)] and National Key Research and Development Program of China [2021YFC2700700 (S.M.C.)]. We also thank the support of the start-up funds of Wuhan University and the National Youth Talents Plan of China.

## Author contributions

S.M.C., W.J.N. and Q.Q.W. conceived and designed the experiments. W.J.N. performed most of the experiments. J.S. conducted the density functional theory calculations. M.R.C., M.G. helped in the data analysis. W.J.N., S.M.C. and Q.Q.W. co-wrote the paper. All authors discussed the results and commented on the manuscript. S.M.C. supervised the overall research.

## Competing interests

The authors declare no competing interests.
