## [Peer Review File · Nature Communications]

Ultra-High-Throughput Mapping of the Chemical Space of Asymmetric Catalysis Enables Accelerated Reaction DiscoveryReviewers' Comments:

Reviewer #1:

Remarks to the Author:

Comments to the Author:

I have read the paper by Chen and co-workers with interest. The work describes their efforts to develop an ion mobility-mass spectrometry (IM-MS) combined with the diastereoisomerization strategy to accelerate the discovery of asymmetric catalytic reactions. Their interest is to perform two asymmetric reactions: the direct asymmetric α -alkylation of aldehydes and the Noyori asymmetric transfer hydrogenation reaction, and they introduce copper (I)-catalyzed azide-alkyne cycloaddition (CuAAC) for fast derivatization. Their analysis of the crude mixtures for ee determination was subsequently assisted by an IM-MS method. Although the authors claim that this general strategy described could be useful in analyzing asymmetric reaction studies, but I am not convinced that it is easier or more precise than chiral chromatography. To me, it appears to be an over complication to use this IM-MS-based approach and I am in doubt whether it is adaptable for the organic chemist without assistance from MS specialists. The IM-MS method cannot directly analyze enantiomers, so further derivatization (e.g., CuAAC) to diastereomers is necessary, and the yield and efficiency of derivatization will affect the analysis results. In particular, IM-MS-based analysis does not allow rapid assessment of the yield of symmetric catalytic reactions. Therefore, I think that one could directly analyze the mixtures using high-performance liquid chromatography with a chiral stationary phase. Overall, all the chemistry depicted in the manuscript were known and the main conclusion is that they perform an IM-MS-based approach for the high-throughput and large-scale screening to optimize asymmetric reactions. I think this work should be published in good analytic journal, but to me it does not reach a level of novelty justifying Nature Communications.

Reviewer #2:

Remarks to the Author:

Agency: NCOMMS-23-21342

Title : "Ultra-High-Throughput Mapping the Chemical Space of Asymmetric Catalysis Enables Accelerated Reaction Discovery"

Author(s): Chen et al

This is a very nice paper about ultra high-throughput screening of enantioselective, catalytic reactions. The method is very new, and relies on clever ion-mobility mass spec techniques. The key is the in situ derivatization of reaction products using click chemistry, to afford diastereomers that have different mobilities in the MS part of the protocol. Of course, this introduces some constraints such as the requirement for an alkyne in one of the reaction partners. A second issue, although the authors don't seem to encounter problems, is the possible kinetic resolution that could occur during the click step, which could cloud the proxy results for ee determination. But, as notes, it seems to be a small effect if it operates at all. Perhaps the authors could make more of this point in their revision.

It is also a nice choice to demonstrate the method with technically demanding reactions, such as a photochemical asymmetric reaction and a ketone hydrogenation. While the "new" catalyst that is discovered for the photochemical alkylation is not that novel, nor is the reaction that exciting given its advanced stage of development, it is still an excellent proof of principle. The authors also engage in optimization studies with their new tools, and this work is well done.

The citations miss some of the early work in the field, and while that is unfortunate, it is certainly fixable. Just a few examples of the work of Miller and Anslyn are shown below. Some consideration of these alternative techniques should probably be worked into the paper to add context and depth.

"Fluorescence-Based Screening of Asymmetric Acylation Catalysts Through Parallel Enantiomer Analysis. Identification of a Catalyst for Tertiary Alcohol Resolution"
Jarvo, E. R.; Evans, C. A.; Copeland, G. T.; Miller, S. J. *J. Org. Chem.* 2001, 66, 5522-5527.

"Selection of Enantioselective Acyl Transfer Catalysts from a Pooled Peptide Library Through a Fluorescence-Based Activity Assay: An Approach to Kinetic Resolution of Secondary Alcohols of Broad Structural Scope"
Copeland, G. T.; Miller, S. J. *J. Am. Chem. Soc.* 2001, 123, 6496-6502.

"From Substituent Effects to Applications: Enhancing the Optical response of a Four-Component Assembly for Reporting EE Values"
Lin, C.-Y.; Giuliano, M. W.; Miller, S. J.; Anslyn, E. V. *Chem. Sci.* 2016, 7, 4085-4090.

Reviewer #3:

Remarks to the Author:

The catalytic and enantioselective transformation is very important in the field of organic synthesis. However, the identification of an efficient condition system faces great challenges due to the complex multidimensional chemical space and low throughput to determine enantiomeric excess of products. This paper shown a high throughput strategy to analyze the enantiomeric excess with combination of ion mobility-mass spectrometry and late-stage diastereoisomerization. Besides, this technology was applied to map the chemical space of asymmetric α -alkylation of aldehydes merged photoredox catalysis and organocatalysis. Then, a novel primary amine chiral catalyst was identified very quickly based on this platform. The research results in this paper were benefit for the study of HTS and data science. This paper could be accepted after minor revision.

- 1) Ion suppression is common in MS and effect the results of quantitative analysis. How did you minimize this effect in this research? Please discuss it in your manuscript and list the details in supporting information.
- 2) All results of asymmetric α -alkylation of aldehydes were only shown in the form of heatmap. Please upload the raw data with supporting information.
- 3) Some detail should be included in Supporting information. Such as the software name and parameter setting, which was used to fit curve when two peaks were overlapped; the concentration of injection, how did you control the temperature of your high throughput experiments, the work-up of the reaction and so on.
- 4) Some expression was confused and should be checked. Such as, Line 23, " of possible combinations among catalysts, substrates, additives, and..." Line 25, "This issue makes the screening of catalysts and their generalities....."
- 5) Minor errors were existed, please check the manuscript and supporting information carefully. For example, page 8 in supporting information , "where A is the $1/K_0$ value", However, A is not existed in equation.

Response to Reviewers' Comments

Reviewer 1:

1. I have read the paper by Chen and co-workers with interest. The work describes their efforts to develop an ion mobility-mass spectrometry (IM-MS) combined with the diastereoisomerization strategy to accelerate the discovery of asymmetric catalytic reactions. Their interest is to perform two asymmetric reactions: the direct asymmetric α -alkylation of aldehydes and the Noyori asymmetric transfer hydrogenation reaction, and they introduce copper (I)-catalyzed azide-alkyne cycloaddition (CuAAC) for fast derivatization. Their analysis of the crude mixtures for ee determination was subsequently assisted by an IM-MS method.

Reply: We appreciate the reviewer for the recognition to our work.

2. Although the authors claim that this general strategy described could be useful in analyzing asymmetric reaction studies, but I am not convinced that it is easier or more precise than chiral chromatography. To me, it appears to be an over complication to use this IM-MS-based approach and I am in doubt whether it is adaptable for the organic chemist without assistance from MS specialists.

Reply: We thank the reviewer for the valuable suggestions to our work.

The large number of possible combinations between catalysts, substrates, additives and reaction conditions constitutes a vast chemical space for asymmetric catalysis, and even modest structural changes to any or a few of these variables can have a profound effect on the experimental outcome. High-throughput screening (HTS) has become the critical approach for revealing the dark space of the reaction reactivities, however, the throughput of asymmetric reaction screening is bottlenecked by the determination of the enantiomeric excess (*ee*) values.

Chiral chromatography such as chiral high-performance liquid chromatography (HPLC) is a mainstream method for the determination of *ee* values of chiral compounds in asymmetric synthesis (*J. Chromatogr. A* **2021**, 1638, 461878). However, it still has some inherent disadvantages. For example, the reaction solution needs to be purified before analysis. For products with new structures, synthesized racemic standards are often required for reference, and a great deal of time is required to select chiral stationary phases suitable for separation and to optimize the separation conditions (*Chem. Soc. Rev.* **2008**, 37, 2593-2608). More importantly, the HPLC separation time for each sample is usually dozens of minutes, although some compounds require shorter separation times. All of these problems severely limit the throughput of chiral chromatography. When analyzing a small number of samples, these lengthy steps are acceptable. However, when performing high throughput screening, conventional chiral chromatography analysis can no longer meet the requirements. For example, to analyze more than 1,000 reactions, chiral HPLC usually takes more than half a month. In contrast, the IM-MS-based method we developed takes only 1.5 minutes to analyze each reaction. For the analysis of the same 1000 reactions, it only takes about 1 day, and the advantage is very significant. The results of our study also confirm that the IM-MS-based method has comparable accuracy to chiral HPLC (Fig. 2F and 2G in the manuscript). In addition, during the mass spectrometry analysis, automated injection and analysis steps are used. In the future, we can automate the data analysis procedure as well, which will allow organic chemists to use this high-throughput analysis method without the assistance of mass spectrometry experts.

3. The IM-MS method cannot directly analyze enantiomers, so further derivatization (e.g., CuAAC) to diastereomers is necessary, and the yield and efficiency of derivatization will affect the analysis results. In particular, IM-MS-based analysis does not allow rapid assessment of the yield of asymmetric catalytic reactions.

Therefore, I think that one could directly analyze the mixtures using high-performance liquid chromatography with a chiral stationary phase.

Reply: We understand the reviewers' concerns. Since the screening of catalytic systems usually proceeds with representative transformations using model substrates, we envision that if a derivatizable substrate is used in the reaction, the enantiomeric products formed could be transformed into the diastereomers by introducing a chiral resolving reagent through in situ derivatization. In this study, CuAAC reaction was used to rapidly convert the enantiomeric reaction products to diastereoisomers thus being analyzed by ion mobility mass spectrometry to achieve high throughput screening. The conversion rate of this click reaction can reach more than 90% in a short period of time (see Fig. S9A in Supplementary Information). The average time cost of derivatization is only a few seconds for one reaction and has little impact on throughput when screening a large number of reactions. It is worth noting that the yield of the derivatization reaction does not affect the accuracy of the *ee* assay because the designed chiral reagents have the same derivatization efficiency for both *R*- and *S*-configuration products. Our experimental results also confirmed the high accuracy of this IM-MS-based method for *ee* determination (see Fig. 2E&2F in the manuscript).

In addition, by adding internal standards that can also be derivatized by chiral reagents to the reaction solution, we can compare the relative yields of the individual reactions on a large scale, thus providing another dimension of important information for the evaluation of the reaction performance, which is not easily available from conventional chiral HPLC analysis (Fig. S9). Of course, the greatest advantage of this method over chiral HPLC is still the ability to rapidly analyze large-scale asymmetric synthesis reactions.

4. Overall, all the chemistry depicted in the manuscript were known and the main conclusion is that they perform an IM-MS-based approach for the high-throughput and large-scale screening to optimize asymmetric reactions.

Reply: We apologize for not being clear about the new chemistry we've discovered. In this study, a new IM-MS-based approach was developed for the high-throughput and large-scale screening of asymmetric reactions. Using this efficient approach, a new class of high-enantioselectivity primary amine organocatalysts of 1,2-diphenylethane-1,2-diamine-based sulfonamides for the organocatalytic and photocatalytic synergistic α -asymmetric alkylation of aldehydes has been discovered by accelerated screening, and the mechanism for the high-selectivity has been demonstrated by computational chemistry. We think that these are new chemistries that will provide new insights into the asymmetric organocatalysis.

Although great progress has been made in asymmetric organocatalysis, catalysts capable of catalyzing the enantioselective conversion of aldehydes to yield highly selective chiral products are still mainly limited to secondary amines, such as imidazolidinone catalyst and pyrrolidine-type secondary amine catalysts (*J. Am. Chem. Soc.* **2009**, *131*, 10875–10877; *J. Am. Chem. Soc.* **2010**, *132*, 13600–13603; *ACS Catal.* **2017**, *7*, 7008–7013). Whether other types of highly enantioselective catalysts (e.g., primary amines) exist is still unknown. Primary amine-based catalysts were used for enantioselective synthesis in some studies, but they were rarely effective for asymmetric α -alkylation reactions of simple aldehydes. By using the high-throughput approach, we indeed discovered a new type of primary amine organocatalysts of 1,2-diphenylethane-1,2-diamine-based sulfonamides for asymmetric α -alkylation of aldehydes with more than 90% *ee* values (Fig. 6A and 6B in manuscript). This new kind of primary amine-based organocatalysts would enrich the chiral amine toolbox in asymmetric organocatalysis.

In addition, the underlying mechanism of the superior enantioselectivity of the sulfonamide-containing organocatalysts was also investigated by mass spectrometry and DFT-based computational chemistry (Fig. 6C-6E). The result indicates the dual hydrogen bonds (N–H \cdots N and N–H \cdots O) between sulfamide group and enamine group in enamine intermediate could be formed, which may assist the formation of *Si* face exposed for the

enantioselective radical addition. This interaction may explain the specific role of sulfonamide group in the catalysts for the high enantioselectivity.

Reviewer 2:

1. This is a very nice paper about ultra high-throughput screening of enantioselective, catalytic reactions. The method is very new, and relies on clever ion-mobility mass spec techniques. The key is the in situ derivatization of reaction projects using click chemistry, to afford diastereomers that have different mobilities in the MS part of the protocol. Of course, this introduces some constraints such as the requirement for an alkyne in one of the reaction partners. A second issue, although the authors don't seem to encounter problems, is the possible kinetic resolution that could occur during the click step, which could cloud the proxy results for ee determination. But, as notes, it seems to be a small effect if it operates at all. Perhaps the authors could make more of this point in their revision.

Reply: We thank the reviewer for the recognition and valuable suggestions to our work, which has strengthened this work. As the reviewer is concerned about the possible kinetic resolution that could occur during the click step, which may cloud the proxy results for ee determination. Actually, we considered the effect of kinetic resolution before the selecting appropriate derivatization reaction and chiral resolving reagent. To minimize the possible effect of kinetic resolution, we selected the click reaction as the optimal derivatization reaction, and the structure of the chiral resolving reagent was specially designed and synthesized. On the one hand, the high chemoselectivity between alkyl group and azide group make the chiral resolving reagent have almost equal reactivity towards enantiomer analytes. On the other hand, the azide group is far away from the chiral center of the chiral resolving reagent, which could minimize the chiral effect during the click step and reduce the kinetic resolution of the chiral resolving reagent to analytes with different configurations. Thus, the unique reactivity between alkyl group and azide group and the suitable distance between chiral center and azide group make chiral resolving reagent show no preference for R- and S- configurations during the post-derivatization procedure. We found that the proposed IM-MS-based method was highly consistent with the quantitative results of chiral HPLC, confirming the accuracy of the developed method (please see Fig. 2F and 2G in the manuscript).

The corresponding discussion has been included in the revised manuscript. See the top of page 5.

2. It is also a nice choice to demonstrate the method with technically demanding reactions, such as a photochemical asymmetric reaction and a ketone hydrogenation. While the “new” catalyst that is discovered for the photochemical alkylation is not that novel, nor is the reaction that exciting given its advanced stage of development, it is still an excellent proof of principle. The authors also engage in optimization studies with their new tools, and this work is well done.

Reply: We thank the recognition of the reviewer to our work.

3. The citations miss some of the early work in the field, and while that is unfortunate, it is certainly fixable. Just a few examples of the work of Miller and Anslyn are shown below. Some consideration of these alternative techniques should probably be worked into the paper to add context and depth.

“Fluorescence-Based Screening of Asymmetric Acylation Catalysts Through Parallel Enantiomer Analysis. Identification of a Catalyst for Tertiary Alcohol Resolution”

Jarvo, E. R.; Evans, C. A.; Copeland, G. T.; Miller, S. J. *J. Org. Chem.* 2001, 66, 5522-5527.

“Selection of Enantioselective Acyl Transfer Catalysts from a Pooled Peptide Library Through a Fluorescence-Based Activity Assay: An Approach to Kinetic Resolution of Secondary Alcohols of Broad Structural Scope”

Copeland, G. T.; Miller, S. J. *J. Am. Chem. Soc.* 2001, 123, 6496-6502.

“From Substituent Effects to Applications: Enhancing the Optical response of a Four-Component Assembly for Reporting EE Values”

Lin, C.-Y.; Giuliano, M. W.; Miller, S. J.; Anslyn, E. V. *Chem. Sci.* 2016, 7, 4085-4090.

Reply: We thank the suggestion of the reviewer, and the citations have been added in the manuscript. Please see refs 15, 16, and 17. The corresponding description in the introduction has been modified.

Reviewer 3:

1. The catalytic and enantioselective transformation is very important in the field of organic synthesis. However, the identification of an efficient condition system faces great challenges due to the complex multidimensional chemical space and low throughput to determine enantiomeric excess of products. This paper shown a high throughput strategy to analyze the enantiomeric excess with combination of ion mobility-mass spectrometry and late-stage diastereoisomerization. Besides, this technology was applied to map the chemical space of asymmetric α -alkylation of aldehydes merged photoredox catalysis and organocatalysis. Then, a novel primary amine chiral catalyst was identified very quickly based on this platform. The research results in this paper were benefit for the study of HTS and data science. This paper could be accepted after minor revision.

Reply: We thank the reviewer for careful reading and constructive comments, which have strengthened this work.

2. Ion suppression is common in MS and effect the results of quantitative analysis. How did you minimize this effect in this research? Please discuss it in your manuscript and list the details in supporting information.

Reply: We agree with the reviewer that the ion suppression is common in electrospray ionization MS using direct infusion manner. However, this effect has little or no impact on the accuracy of the quantification of the *ee* values. In our method, the *ee* values are calculated based on the ratio of MS peak intensities between the diastereoisomeric products generated after derivatization. Since the ionization efficiencies between diastereoisomers are almost the same, they suffer from the equivalent ion suppression effect. Therefore, the ion suppression does not affect the accuracy of the ratio between them, and thus does not have an impact on the accuracy of the *ee* value determination. Nevertheless, the ion suppression effect may have some influence on the detection sensitivity of the derivatized products. Therefore, we added sodium chloride solution (100 μ M) to the detection system to enhance the MS signal intensity of the sodium-adducted target product ions.

In addition, the ion suppression effect is likely to have some impact on the determination of relative yields because we used only one internal standard to correct for products with different substituents, and the ionization efficiencies may differ slightly between products. To reduce the influence of this effect, we used the protonated MS peaks for quantification. This is because both the product and the internal standard derived from the click reaction contain a triazole structure, which has a high and similar protonation efficiency in different compounds. Therefore, the ionization difference can be minimized to increase the accuracy of quantification.

In summary, although there is an ion suppression effect during mass spectrometry detection, we minimize the effect of ion suppression on quantitative results by taking appropriate measures. The effect of ion suppression

has been discussed and modified as the reviewer's suggestion. Please see page 13 in the manuscript and page 3 in Supplementary Information.

2) All results of asymmetric α -alkylation of aldehydes were only shown in the form of heatmap. Please upload the raw data with supporting information.

Reply: The raw data of the results of asymmetric α -alkylation of aldehydes was uploaded as a source data.

3) Some detail should be included in Supporting information. Such as the software name and parameter setting, which was used to fit curve when two peaks were overlapped; the concentration of injection, how did you control the temperature of your high throughput experiments, the work-up of the reaction and so on.

Reply: We thank the suggestions of the reviewer, and these details were added in supporting information on page 5.

4) Some expression was confused and should be checked. Such as, Line 23, “ of possible combinations among catalysts, substrates, additives, and” Line 25, “ This issue makes the screening of catalysts and their generalities.....”

Reply: We apologize for the confusion. The corresponding expression have been modified in the manuscript.

5) Minor errors were existed, please check the manuscript and supporting information carefully. For example, page 8 in supporting information, “where A is the $1/K_0$ value” , However, A is not existed in equation.

Reply: We thank the reviewer for patiently reading, and we have checked and modified the manuscript and Supplementary Information.

Reviewers' Comments:

Reviewer #1:

Remarks to the Author:

The revised manuscript entitled "Ultra-High-Throughput Mapping the Chemical Space of Asymmetric Catalysis Enables Accelerated Reaction Discovery" reports an ion mobility-mass spectrometry (IM-MS) combined with the diastereoisomerization strategy to accelerate the discovery of asymmetric catalytic reactions. The authors have shown a lot of effort to address my concerns (derivatization and novelty issues) carefully by a response letter. I personally agree with the author's claim that this method allows rapid screening of catalysts and ligands for asymmetric reactions to improve the traditional use of chiral high-performance liquid chromatography (HPLC) methods. In addition, I recommend that the authors revise a few additional points listed below.

In SI, page 24, compound 3d: ..."1.93 (t, = 1.8, 1H)" should be revised to "1.93 (t, J = 1.8 Hz, 1H)".

In SI, pages 27 and 29, compound A26 and compound A31: ..."13C NMR (151 MHz, CDCl3)" should be revised to "13C NMR (151 MHz, CDCl3)".

---Please check the NMR data of the compounds:

In SI, page 29, compound A32: ..."6.82 (m, J = 8.8, 3.6, 1.4 Hz, 2H)", "4.51 (m, J = 6.1 Hz, 1H)",...

In SI, page 30, compound A34: ..."7.58 (td, J = 8.2, 6.8, 1.3 Hz, 1H)", "7.52 (td, J = 8.1, 6.8, 1.3 Hz, 1H)",...

Reviewer #2:

Remarks to the Author:

I am satisfied with the changes and recommend publication.

Reviewer #3:

Remarks to the Author:

The authors addressed my concerns well and I recommend to accept it for publication.

Response to Reviewers' Comments

Reviewer 1:

1. The revised manuscript entitled "Ultra-High-Throughput Mapping the Chemical Space of Asymmetric Catalysis Enables Accelerated Reaction Discovery" reports an ion mobility-mass spectrometry (IM-MS) combined with the diastereoisomerization strategy to accelerate the discovery of asymmetric catalytic reactions. The authors have shown a lot of effort to address my concerns (derivatization and novelty issues) carefully by a response letter. I personally agree with the author's claim that this method allows rapid screening of catalysts and ligands for asymmetric reactions to improve the traditional use of chiral high-performance liquid chromatography (HPLC) methods.

Reply: We appreciate the reviewer for the recognition to our work.

2. In addition, I recommend that the authors revise a few additional points listed below.

In SI, page 24, compound 3d: ..."1.93 (t, $J = 1.8$, 1H)" should be revised to "1.93 (t, $J = 1.8$ Hz, 1H)".

In SI, pages 27 and 29, compound A26 and compound A31: ..."13C NMR (151 MHz, CDCl₃)" should be revised to "13C NMR (151 MHz, CDCl₃)".

---Please check the NMR data of the compounds:

In SI, page 29, compound A32: ..."6.82 (m, $J = 8.8, 3.6, 1.4$ Hz, 2H)", "4.51 (m, $J = 6.1$ Hz, 1H)",...

In SI, page 30, compound A34: ..."7.58 (td, $J = 8.2, 6.8, 1.3$ Hz, 1H)", "7.52 (td, $J = 8.1, 6.8, 1.3$ Hz, 1H)",...

Reply: We thank the reviewer for the valuable suggestions to our work. All the mistakes have been corrected in the Supplementary Information.

Reviewer 2:

1. I am satisfied with the changes and recommend publication.

Reply: We thank the reviewer for the recognition and valuable suggestions to our work.

Reviewer 3:

1. The authors addressed my concerns well and I recommend to accept it for publication.

Reply: We thank the reviewer for careful reading and constructive comments, which have strengthened this work.